# Crop Production and Carbon Sequestration Potential of *Grewia oppositifolia*-Based Traditional Agroforestry Systems in Indian Himalayan Region

Naveen Tariyal [1], Arvind Bijalwan [1], Sumit Chaudhary [1], Bhupendra Singh [1], Chatar Singh Dhanai [1], Sumit Tewari [2], Munesh Kumar [3], Sandeep Kumar [1], Marina M. S. Cabral Pinto [4] and Tarun Kumar Thakur [5,*]

1    College of Forestry, Veer Chandra Singh Garhwali Uttarakhand University of Horticulture and Forestry, Ranichauri 249199, India; naveentariyal3@gmail.com (N.T.); arvindbijalwan276@gmail.com (A.B.); sumit38019@gmail.com (S.C.); butola_bs@yahoo.co.in (B.S.); dhanaiagro@gmail.com (C.S.D.); sandeepprabhakar1@gmail.com (S.K.)
2    Forest Research Institute, Dehradun 248006, India; tewaris.439@gmail.com
3    Department of Forestry and Natural Resources, HNB Garhwal University, Srinagar Garhwal 249161, India; muneshmzu@yahoo.com
4    GeoBioTec Research Centre, Department of Geosciences, University of Aveiro, 3810-193 Aveiro, Portugal; marinacp@ua.pt
5    Department of Environmental Science, Indira Gandhi National Tribal University, Amarkantak 484887, India
*    Correspondence: tarun.thakur@igntu.ac.in

**Abstract:** Bhimal (*Grewia oppositifolia*) is the most important agroforestry tree species used for fodder, fuel and fiber in the Himalayan region. In the present study, *G. oppositifolia*-based traditional agroforestry systems were selected for the estimation of carbon stock and the production potential of barnyard millet (*Echinochloa frumentacea*) and finger millet (*Eleusine coracana*), with two elevational ranges, i.e., 1000–1400 and 1400–1800 m amsl, in Garhwal Himalaya, India. The results of the investigation showed a decline in the growth and yield attributes of both the millet crops under the *G. oppositifolia*-based agroforestry system at both elevations as compared to their respective control sites (sole crops). Among the elevations, the total number of tillers per plant (2.70 and 2.48), the number of active tillers per plant (2.18 and 2.25), panicle length (17.63 cm and 6.95 cm), 1000-seed weight (5.49 g and 4.33 g), grain yield (10.77 q ha$^{-1}$ and 11.35 q ha$^{-1}$), straw yield (37.43 q ha$^{-1}$ and 30.15 q ha$^{-1}$), biological yield (48.21 q ha$^{-1}$ and 41.51 q ha$^{-1}$) and the harvest index (22.53% and 27.78%) were recorded as higher in the lower elevation in both *E. frumentacea* and *E. coracana*, respectively, while plant population per m$^2$ (18.64 and 25.26, respectively) was recorded as higher in the upper elevation. Plant height for *E. frumentacea* (180.40 cm) was also observed to be higher in the upper elevation, while for *E. coracana* (98.04 cm), it was recorded as higher in the lower elevation. Tree carbon stock was reported negatively with an increase in altitude. The maximum amount of sequestered carbon in the tree biomass for *G. oppositifolia* was 23.29 Mg ha$^{-1}$ at the lower elevation and 18.09 Mg ha$^{-1}$ at the upper elevation. Total carbon stock in the tree biomass was reported to be the highest (15.15 Mg ha$^{-1}$) in the 10–20 cm diameter class, followed by 20–30 cm (6.99 Mg ha$^{-1}$), >30 cm (2.75 Mg ha$^{-1}$) and the lowest (2.32 Mg ha$^{-1}$) in the <10 cm diameter class. The results show that the yield of *E. frumentacea* and *E. coracana* was not reduced so severely under the *G. oppositifolia* system; however, keeping in mind the other benefits of this multipurpose tree, i.e., carbon sequestration and socioecological relevance, farmers can get benefit from adopting these crops under *G. oppositifolia*-based agroforestry systems.

**Keywords:** millet; intercropping; elevation; crop productivity; biomass; carbon stock

## 1. Introduction

Agroforestry is a land management technique that uses woody perennial crops and/or animals on farms to promote productivity, diversity, and long-term production, and improved economic and environmental benefits at various stages to the land users [1,2]. Agroforestry is helpful in enhancing tree cover and reducing stress on natural forests and is the most preferred land-use system for improving and restoring unproductive lands [3]. Agroforestry has been seen as a holistic approach during recent years because of the rising appreciation of the importance of the trees outside the forest. In India, they are a major source of various products, i.e., fodder, timber, fuelwood and other miscellaneous supplies derived from the agroforestry area [4]. Carbon sequestration through tree–crop combination works as an effective tool to manage the excess amount of carbon and reduce the level of greenhouse gases through biomass production, and the plantation of slow-growing trees effectively increases the carbon stock over a longer period [5].

Forests have been recognized as significant carbon sinks; their role in lowering $CO_2$ emissions and improving carbon sinks is widely accepted. Over the years, agroforestry has been recognized as a suitable way to reduce the level of atmospheric $CO_2$ through above and below ground carbon sequestration in trees and crop biomass, as well as in soil ecosystems, as compared to conventional agriculture [6]. Climate change has been a very big problem for the past few decades and requires a long-lasting solution for mitigation throughout the world [7]. Agroforestry has the potential to improve the resilience of the ecosystem and cope with the negative impacts of climate change [8]. Oelbermann et al. [9] found promising results of carbon sequestration in tropical and temperate agroforestry systems in Costa Rica and southern Canada. Agroforestry systems in Southern Europe act as significant carbon sinks and also reduces forest fires and carbon inputs in the atmosphere [10]. Agroforestry systems in Africa, such as home gardens, live fences and parklands, possess substantial carbon stocks and accumulate 0.2–0.8 Mg C ha$^{-1}$ per year [11]. India has a long tradition of practicing agroforestry, which includes trees on farmlands, community forests and various forest management and ethno-forestry practices [12]. In India, farmers have an old practice of growing trees and retaining scattered trees on farmland, which has not been changed much over the centuries. These trees are mainly multipurpose and used for fruits, fodder, fuelwood, shade and medicinal purposes.

The Garhwal Himalaya of Uttarakhand, situated in the northern part of India is equipped with many traditional agroforestry systems, dominated by *Grewia oppositifolia*, *Celtis australis* and *Quercus leucotrichophora*; these systems have been present from time immemorial. The farmers have been growing many seasonal, biennial and perennial crops under these systems. The hilly regions of Uttarakhand are dominated by various agrisilviculture, agrihorticulture and agrisilvihorticulture systems. In agrisilviculture system, the agriculture crops, *viz.*, *Echinochloa frumentaceae*, *Eleusine coracana*, *Triticum aestivum*, *Amaranthus* spp., *Vigna umbellata* and *Oryza sativa* are grown along with tree species such as *Grewia oppositifolia*, *Melia azedarach* and *Celtis australis*. Tree–crop combinations also change with the elevation [13]. In agroforestry, the interaction of trees and crops and the microclimate also affects the productivity of growing crops [14].

*G. oppositifolia,* (synonym: *Grewia optiva*) locally known as bhimal, bihul, biung and dhaman, is a sub-tropical tree species and belongs to the family Malvaceae. It is one of the important multipurpose agroforestry tree species and dominates farmers' fields in the lower and mid-hills of the Himalayas [15]. It provides green and protein-rich fodder, especially during the lean (winter) period [16], and also provides fiber, fuel and various other services such as carbon sequestration, purifying oxygen, improving agrobiodiversity and reducing soil erosion [17]. Traditionally, tender twigs of *G. oppositifolia* have been used for making baskets and other domestic products [18]. The climatic conditions and topography of the hilly districts of Uttarakhand have good potential for millet production from time immemorial [19]. Small millets are grown in areas where the climatic and edaphic conditions are inappropriate for rice or other high-value crops to thrive [20]. *E. frumentaceae* and *E. coracana* are grown mainly for human dietary purposes and as fodder for livestock.

These crops are very nutritious and a great source of carbohydrates, proteins and fibers, and are among the famous "*Barahnaja*" crop of Garhwal [21]. Productivity of pure or sole agriculture crops is always high due to a lack of interference of other components and heavy investment in agrochemicals and fertilizers for promised food security. On the other hand, challenges such as nature conservation, deterioration of soil health, scarcity of water resources, poor air quality and ecological imbalance caused by intensive agriculture practices in India are posing a severe threat to the long-term viability of agro-ecosystems. Agroforestry could be a solution to the problem because it can sustain productivity as well as ecological balance for a longer period [22]. Agroforestry systems play a critical role in the sustenance of production and the productivity of farmland, giving rise to a resilient farming system, and thereby enhancing livelihoods and employment opportunities. They also help in the quality restoration of degraded land and carbon sequestration [1]. Keeping in view the aforementioned points, the present study was aimed to estimate the carbon sequestration potential of a *G. oppositifolia*-based traditional agroforestry system and the productivity of *E. frumentaceae* and *E. coracana* under this system in two broad elevation ranges in Garhwal Himalaya, India.

## 2. Materials and Methods

### 2.1. Study Area

The study was carried out in *G. oppositifolia*-based traditional agroforestry systems (Figure 1) in six villages of the Tehri Garhwal district, Uttarakhand, India, with the two elevational ranges, i.e., 1000–1400 m amsl (lower elevation) and 1400–1800 m amsl (upper elevation), under which *E. frumentaceae* and *E. coracana* were cultivated. Three villages (Devri, Pali and Bhali) were selected for study in the lower elevation and three villages (Moun, Kotdwara and Kainchhu) in the upper elevation. The geographical information of the study sites is presented in Table 1 and the location map is depicted in Figure 2.

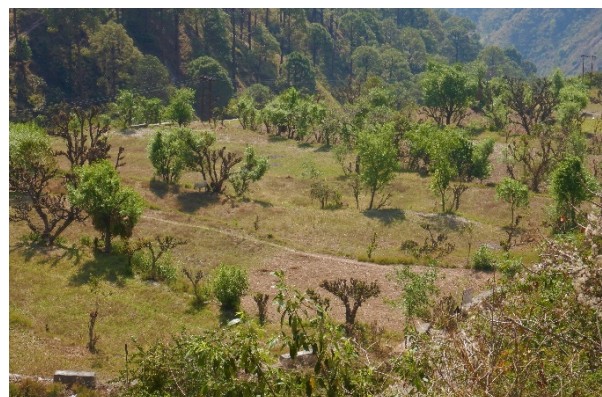 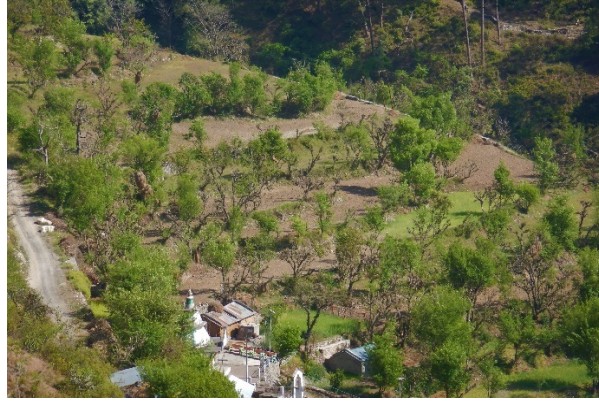

**Figure 1.** Overview of *Grewia oppositifolia*-based traditional agroforestry system in Garhwal Himalaya.

**Table 1.** Geographic information (co-ordinates) of study sites.

| Elevation Range (m amsl) | Site | Latitude (N) | Longitude (E) |
|---|---|---|---|
| Lower elevation (1000–1400) | Devri | 30°21′04.07″ | 78°22′15.36″ |
| | Pali | 30°15′46.91″ | 78°25′45.96″ |
| | Bhali | 30°15′37.32″ | 78°25′49.13″ |
| Upper elevation (1400–1800) | Moun | 30°18′08.04″ | 78°23′33.96″ |
| | Kotdwara | 30°17′36.60″ | 78°25′24.72″ |
| | Kainchhu | 30°17′50.03″ | 78°26′12.78″ |

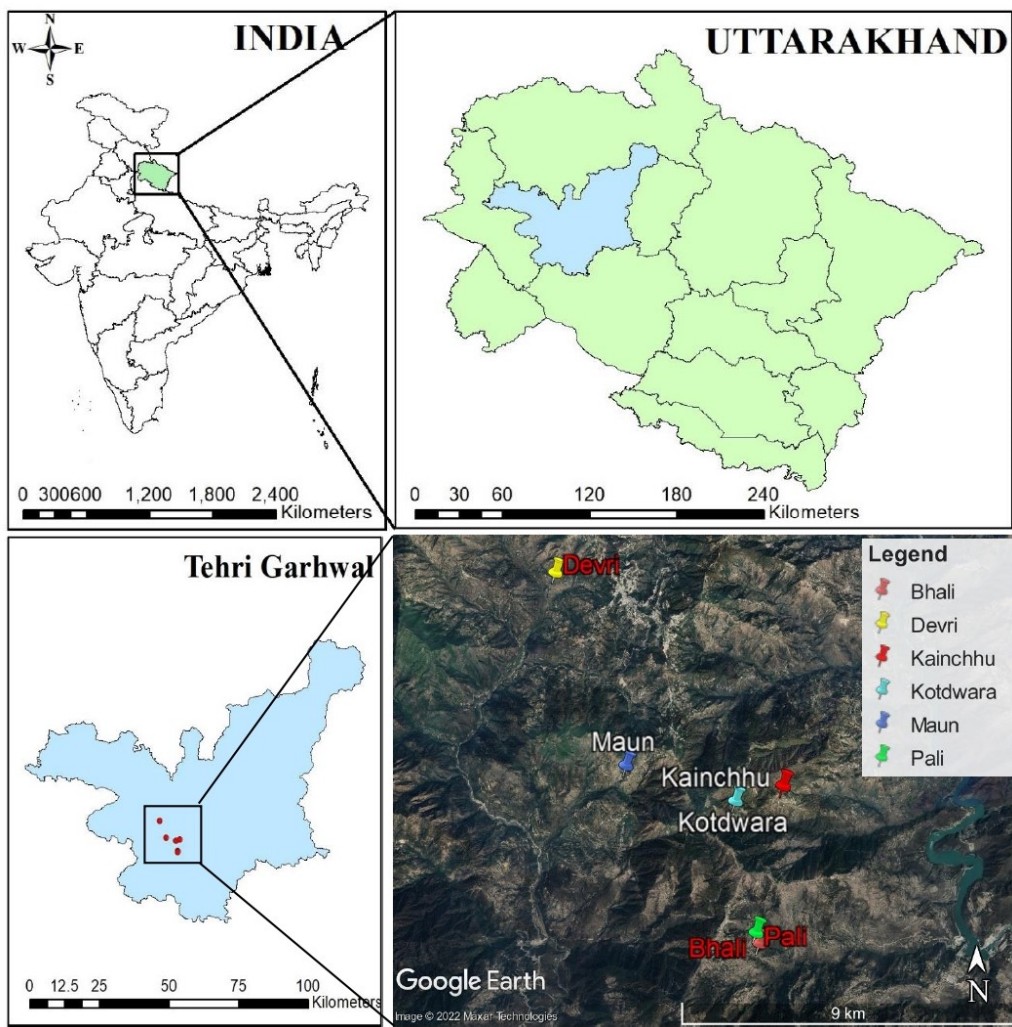

**Figure 2.** The location map of the study area.

### 2.2. Climate and Soil

The climate of the Tehri district ranges from sub-tropical to temperate regions. Normally, the coldest months are December and January, while May and June are the hottest. During summer, the maximum temperature rises to 36 °C, while in winter, the temperature has been recorded in negative values [23]. Snowfall is very common during the winter in temperate areas. The average annual rainfall was recorded between 1200 and 1500 mm, of which the maximum amount is received from June to September. The soil is loam and acidic, and brown to dark brown [23,24]. The soils of the study area are moderate to very shallow, excessively drained, stony and severely eroded due to steep slopes [24].

### 2.3. Experimental Details

The elevation and geographical location of the study sites were measured using a Garmin Vista handheld GPS (Garmin International Inc., Olathe, KS, USA). Stratified random sampling was adopted in the present study to obtain sample plots. Ten sample plots of 0.04 ha (20 × 20 m$^2$) were laid down in both cropping systems in each site for the measurement of tree characteristics, *viz.*, height and dbh (diameter at breast height). Sample plots of 1 × 1 m$^2$ were laid out randomly within the previously studied sample plots (i.e., 0.04 ha) to estimate the growth and yield attributes of *E. frumentaceae* and *E. coracana*, under the *G. oppositifolia*-based agroforestry system (Figure 3) and their respective sole (control) cropping system.

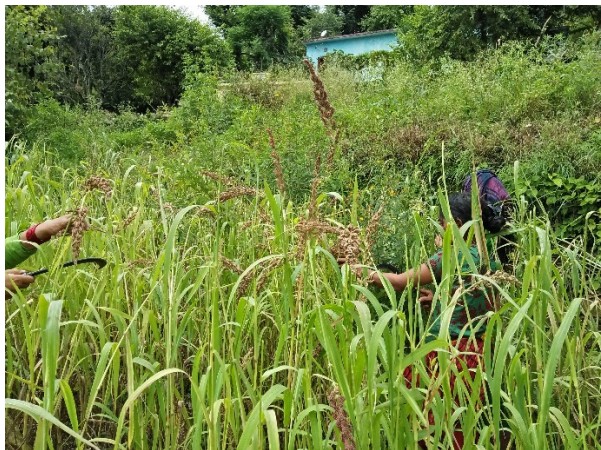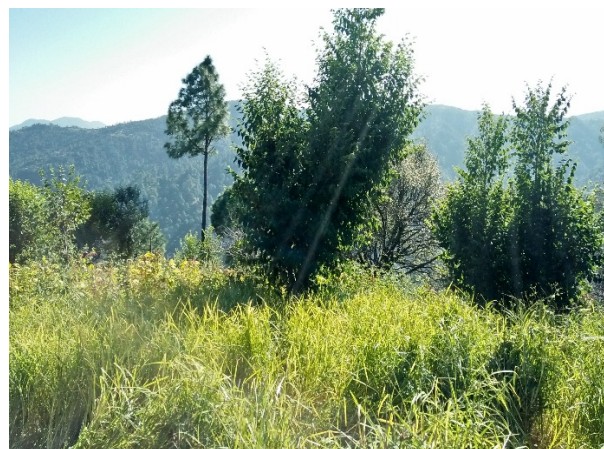

**Figure 3.** Cultivation of *E. frumentaceae* and *E. coracana* under *G. oppositifolia*-based traditional agroforestry system in Garhwal Himalaya.

*2.4. Observed Parameters of Agricultural Crops*

To evaluate the growth and productivity of both selected crops under agroforestry and their respective control condition, the observations for both agricultural crops were recorded during the Kharif season (April–September) for plant population (per m$^2$), plant height (cm), number of total tillers per plant, number of active tillers per plant, panicle length (cm), 1000-grain weight (gm), grain yield (q ha$^{-1}$) and straw yield (q ha$^{-1}$), for both the selected crops. The harvest index (HI) was also calculated using the following equation

$$\text{Harvest index (HI)} = \frac{\text{Economic (grain) yield}}{\text{Biological (grain + straw) yield}} \times 100$$

*2.5. Estimation of Tree Biomass and Carbon Stock*

Tree volumes and carbon stocks were estimated by non-destructive methods using tree dbh (diameter at breast height), i.e., 1.37 m above the ground level, and the height of trees. Tree height and dbh were measured using a Ravi Multimeter and a tree caliper, respectively. The volume of the individual standing tree was calculated with the help of the recorded dbh and height, using the volume equation for the individual tree and the generic volume equation for the Tehri Garhwal region (Table 2), developed by the Forest Survey of India [25]. Trees species were identified using *Flora of the District Garhwal, North West Himalaya* [26].

**Table 2.** Volume equation of different tree species.

| Common Name | Botanical Name | Volume Equation |
| --- | --- | --- |
| Bhimal | *Grewia oppositifolia* | $V = -0.44075 + 7.49221D - 36.09962D^2 + 71.91238D$ |
| Banj | *Quercus leucotrichophora* | $V/D^2 = 0.085356/D^2 - 1.25189/D + 7.702984$ |
| Bedu | *Ficus palmata* | $\sqrt{V} = 0.03629 + 3.95389D - 0.84421\sqrt{D}$ |
| Timla | *Ficus auriculata* | $\sqrt{V} = 0.03629 + 3.95389D - 0.84421\sqrt{D}$ |
| Toon | *Toona ciliata* | $V = 1.10314 - 3.52579\sqrt{D} + 15.50182D^2$ |
| Chir pine | *Pinus roxburghii* | $\sqrt{V} = 0.05131 + 3.9859D - 1.0245\sqrt{D}$ |
| Buransh | *Rhododendron arboreum* | $V = 0.06007 - 0.21874\sqrt{D} + 3.63428D^2$ |
| Rest of Species | – | $V = 0.00855 + 0.4432D^2 + 0.28813D^2H$ |

The Growing Stock Volume Density (GSVD) was used to indicate the sum of the total volume of each tree species within the sample plot, where GSVD was multiplied by 25 to convert it into GSVD m$^3$ ha$^{-1}$. The Suitable Biomass Expansion Factor (BEF) was used for calculating the Above Ground Biomass Density (AGBD) [27].

Hardwood:   BEF = exp{1.91 − 0.34 × ln(GSVD)} (for GSVD ≤ 200 m$^3$ ha$^{-1}$)
            BEF = 1.0 (for GSVD > 200 m$^3$ ha$^{-1}$)
            BEF = 1.68 (for GSVD < 10 m$^3$ ha$^{-1}$)
Pine:       BEF = 0.95 (for GSVD is 10–100 m$^3$ ha$^{-1}$)
            BEF = 0.81 (for GSVD > 100 m$^3$ ha$^{-1}$)

The Below Ground Biomass Density (BGBD) was estimated using the following regression equation, developed by Cairns et al. [28].

$$BGBD = \exp\{-1.059 + 0.884 \times \ln(AGBD) + 0.284\}$$

The total biomass is the summation of the AGBD and the BGBD of all the tree species. The total biomass value was converted into Total Carbon Stock (Mg ha$^{-1}$) using the Intergovernmental Panel on Climate Change fraction, multiplying by the default value of 0.50 C [29].

$$TCD \text{ (Mg ha}^{-1}) = TBD \text{ (Mg ha}^{-1}) \times 0.50$$

*2.6. Statistical Analysis*

A one-way ANOVA was used to analyze the carbon sequestration of trees; data were analyzed using Microsoft Office Excel 2007. Data on the crop growth and yield of *E. frumentaceae* and *E. coracana* were analyzed using the statistical program, "STPR-3", developed by the Department of Mathematics and Statistics, College of Basic Science and Humanities, Govind Ballabh Pant University of Agriculture and Technology, Pantnagar, India. The critical difference (CD) was used to test the level of significance between the treatments at a 5% level of probability.

## 3. Results

*3.1. Growth Attributes of Selected Agricultural Crops*

Plant height, plant population per m$^2$, panicle length, and the number of total tillers and active tillers per plant showed significant ($p = 0.05$) variations among the studied villages for both the millet crops. At both the elevations, all the maximum growth parameters were recorded under the sole cropping (control) system (Table 3). Among the studied villages of the lower elevation, the maximum plant populations were recorded in Devri for both *E. frumentaceae* and *E. coracana* (16.85 and 26.60, respectively), whereas at the upper elevation, the maximum plant population of *E. frumentaceae* was recorded in Kotdwara (19.76), while the maximum plant population of *E. coracana* was reported in Moun (24.98). Among the agroforestry systems of the study sites at the lower elevation, the maximum plant height for *E. frumentaceae* (181.7 cm) was reported in Pali, while the upper elevation maximum plant height (176.3 cm) was obtained in Kainchhu. However, for *E. coracana*, the maximum plant height in the lower elevation was recorded in Bhali (95.04 cm) and at the upper elevation in Kotdwara (98.7 cm). Under the tree-based land use systems, the higher number of total tillers, active tillers per plant and panicle length were recorded in the lower elevation (1000–1400 m amsl) as compared to the upper elevation. For *E. frumentaceae*, the maximum total number of tillers per plant was observed in Devri and Pali (2.40) at the lower elevation, while at the upper elevation, the maximum number of total tillers was observed in Moun (2.00); whereas for finger millet, at the lower elevation, the maximum total tillers was recorded in Bhali (2.62) and at the upper elevation in Kotdwara (2.16). The maximum number of active tillers per plant under agroforestry was reported in Devri (1.87) and Bhali (2.31) at the lower elevations, while at the upper elevation, the maximum number of active tillers per plants was observed in Moun (1.75) and Kotdwara (1.93) for *E. frumentaceae* and *E. coracana*, respectively. The maximum panicle length of *E. frumentaceae* and *E. coracana* under agroforestry were reported in Devri (17.5 cm and 7.11 cm, respectively) at the lower elevation, while at the upper elevation, in Kotdwara (15.8 cm) and Kainchhu (7.08 cm) for *E. frumentaceae* and *E. coracana*, respectively (Table 4). Among the elevations, the growth

and yield attributes, i.e., the number of total tillers per plant (2.70 and 2.48), the number of active tillers per plant (2.18 and 2.25) and panicle length (17.63 cm and 6.95 cm), were recorded as higher at the lower elevation in *E. frumentaceae* and *E. coracana*, respectively, while plant population per m$^2$ (18.64 and 25.26, respectively) was recorded as higher at the upper elevation under the *G. oppositifolia*-based agroforestry system. The average plant height for *E. frumentaceae* (180.40 cm) was also observed as higher in the upper elevation, whereas for *E. coracana* (98.04 cm), it was recorded as higher in the lower elevation (Table 5).

**Table 3.** Growth attributes of *E. frumentacea* and *E. coracana* under *G. oppositifolia*-based agroforestry system.

| Elevation/Site | | Plant Population (per m$^2$) | | Plant Height (cm) | | Total Tiller | | Active Tiller | | Panicle Length (cm) | |
|---|---|---|---|---|---|---|---|---|---|---|---|
| | | *E. frumentacea* | *E. coracana* | *E. frumentacea* | *E. coracana* | *E. frumentacea* | *E. coracana* | *E. frumentacea* | *E. coracana* | *E. frumentacea* | *E. coracana* |
| Lower elevation | Devri | 16.85 | 26.60 | 174.00 | 91.50 | 2.40 | 1.99 | 1.87 | 1.79 | 17.50 | 7.11 |
| | Pali | 14.45 | 20.78 | 181.70 | 94.00 | 2.40 | 2.08 | 1.79 | 1.84 | 16.50 | 6.39 |
| | Bhali | 15.24 | 22.40 | 164.90 | 95.04 | 2.00 | 2.62 | 1.49 | 2.31 | 16.20 | 6.52 |
| | Control (C1) | 20.94 | 27.83 | 198.30 | 111.60 | 4.00 | 3.24 | 3.57 | 3.06 | 20.30 | 7.76 |
| Upper elevation | Moun | 15.80 | 24.98 | 174.90 | 91.40 | 2.00 | 1.93 | 1.75 | 1.73 | 14.90 | 5.92 |
| | Kotdwara | 19.76 | 23.00 | 160.10 | 98.70 | 1.70 | 2.16 | 1.44 | 1.93 | 15.80 | 6.48 |
| | Kainchhu | 17.06 | 23.28 | 176.30 | 93.80 | 1.90 | 1.94 | 1.65 | 1.67 | 15.40 | 7.08 |
| | Control (C2) | 21.93 | 29.79 | 210.30 | 104.70 | 2.60 | 2.89 | 2.31 | 2.64 | 18.30 | 7.30 |
| SEM (±) | | 0.64 | 0.83 | 7.60 | 3.88 | 0.13 | 0.90 | 0.93 | 0.10 | 0.97 | 0.35 |
| CD (*p* = 0.05) | | 1.84 | 2.41 | 22.1 | 11.3 | 0.395 | 0.26 | 0.27 | 0.29 | 2.83 | 1.01 |

**Table 4.** Yield attributes of *E. frumentacea* and *E. coracana* under *G. oppositifolia*-based agroforestry system.

| Elevation/Site | | 1000-Seed Weight (g) | | Grain Yield (q ha$^{-1}$) | | Straw Yield (q ha$^{-1}$) | | Biological Yield (q ha$^{-1}$) | | Harvest Index (%) | |
|---|---|---|---|---|---|---|---|---|---|---|---|
| | | *E. frumentacea* | *E. coracana* | *E. frumentacea* | *E. coracana* | *E. frumentacea* | *E. coracana* | *E. frumentacea* | *E. coracana* | *E. frumentacea* | *E. coracana* |
| Lower elevation | Devri | 5.72 | 4.27 | 10.01 | 10.85 | 32.69 | 28.13 | 42.71 | 38.98 | 23.56 | 27.89 |
| | Pali | 4.72 | 3.90 | 9.74 | 10.46 | 32.48 | 25.62 | 42.22 | 36.08 | 23.04 | 28.86 |
| | Bhali | 5.20 | 4.37 | 9.75 | 11.36 | 33.49 | 26.24 | 43.24 | 37.59 | 22.50 | 30.46 |
| | Control (C1) | 6.32 | 4.78 | 13.59 | 12.77 | 51.09 | 40.64 | 64.68 | 53.41 | 21.00 | 23.90 |
| Upper elevation | Moun | 4.62 | 3.98 | 8.45 | 8.61 | 26.11 | 21.40 | 34.56 | 30.01 | 24.40 | 28.50 |
| | Kotdwara | 4.20 | 3.73 | 8.04 | 8.07 | 26.22 | 22.03 | 34.25 | 30.90 | 23.80 | 26.20 |
| | Kainchhu | 3.77 | 3.57 | 7.54 | 7.30 | 25.12 | 20.66 | 32.35 | 26.96 | 23.12 | 28.20 |
| | Control (C2) | 4.91 | 4.33 | 12.63 | 10.13 | 42.35 | 31.66 | 54.98 | 41.58 | 23.00 | 24.30 |
| SEM (±) | | 0.39 | 0.30 | 0.46 | 0.63 | 1.78 | 1.74 | 1.99 | 2.27 | 1.40 | 1.64 |
| CD (*p* = 0.05) | | 1.15 | 0.88 | 1.33 | 1.82 | 5.15 | 5.04 | 5.77 | 6.58 | 5.48 | 4.77 |

**Table 5.** Average growth and yield attributes of *E. frumentacea* and *E. coracana* at two elevations under *G. oppositifolia*-based agroforestry system.

| Observed Plant Parameter | Echinochloa frumentacea | | Eleusine coracana | |
|---|---|---|---|---|
| | Lower Elevation | Upper Elevation | Lower Elevation | Upper Elevation |
| Plant population (per m$^2$) | 16.87 | 18.64 | 24.40 | 25.26 |
| Total tiller | 2.70 | 2.05 | 2.48 | 2.23 |
| Active tiller | 2.18 | 1.79 | 2.28 | 1.99 |
| Panicle length (cm) | 17.63 | 16.10 | 6.95 | 6.70 |
| Plant height (cm) | 179.73 | 180.40 | 98.04 | 97.15 |
| 1000-seed weight (g) | 5.49 | 4.38 | 4.33 | 3.90 |
| Grain yield (q ha$^{-1}$) | 10.77 | 9.17 | 11.36 | 8.53 |
| Straw yield (q ha$^{-1}$) | 37.44 | 29.95 | 30.16 | 23.94 |
| Biological yield (q ha$^{-1}$) | 48.21 | 39.04 | 41.51 | 32.36 |
| Harvest index (%) | 22.53 | 23.58 | 27.78 | 26.80 |

*3.2. Yield Parameters of Selected Agriculture Crops*

Significant ($p$ = 0.05) variation was also observed for the 1000-seed weight (test weight), straw yield, grain yield and biological yield, for both *E. frumentaceae* and *E. coracana* among the studied sites. At both the elevations, the control (sole crop) recorded the highest values for all the above-mentioned parameters of both millet crops (Table 4). Among the elevations, the yield attributes, *viz.,* the 1000-seed weight (5.49 and 4.33 g), grain yield (10.77 and 11.35 q ha$^{-1}$), straw yield (37.43 and 30.15 q ha$^{-1}$) and biological yield (48.21 and 41.51 q ha$^{-1}$) were recorded as higher in the lower elevation for both *E. frumentaceae* and *E. coracana*, respectively. The means of the growth and yield attributes of both agricultural crops at the lower and upper elevation ranges under a *G. oppositifolia*-based agroforestry system are presented in Table 5. The maximum 1000-seed weight for barnyard and finger millet was recorded in Devri (5.72 g) and Bhali (4.37 g) villages, respectively, at the lower elevation, while in the upper elevation, the maximum 1000-seed weights for both crops were recorded in Moun (4.62 g and 3.98 g), respectively. Maximum grain yield under agroforestry was reported in Devri (10.01 q ha$^{-1}$) and Bhali (11.35 q ha$^{-1}$) at the lower elevation, while at the upper elevation, in Moun (8.45 and 8.61 q ha$^{-1}$, respectively) for *E. frumentaceae* and *E. coracana*. Maximum straw yield for *E. frumentaceae* and *E. coracana* under agroforestry was reported in Bhali (33.49 q ha$^{-1}$) and Devri (28.13 q ha$^{-1}$), respectively, at the lower elevation, while at the upper elevation, in Kotdwara, i.e., 26.21 q ha$^{-1}$ for *E. frumentaceae* and 22.03 q ha$^{-1}$ for *E. coracana*. Among the villages, for *E. frumentaceae*, the maximum biological yield was observed in Bhali (43.23 q ha$^{-1}$) at the lower elevation, while at the upper elevation, in Moun (34.56 q ha$^{-1}$), whereas for *E. coracana* at the lower elevation, the maximum biological yield was recorded in Devri (38.98 q ha$^{-1}$) and at the upper elevation in Kotdwara (30.89 q ha$^{-1}$). The maximum harvest index was reported in Devri (23.56) and Bhali (30.46) at the lower elevation, while at the upper elevation, in Moun (24.40 and 28.50) for *E. frumentaceae* and *E. coracana*, respectively (Table 4). The means of the growth and yield attributes of both agricultural crops at the lower and upper elevation ranges under the *G. oppositifolia*-based agroforestry system are presented in Table 5.

*3.3. Tree Carbon Stock*

In the present study, *G. oppositifolia* was found to be the dominant tree in agroforestry, with the main associated tree species of *Celtis australis* being found in all the studied sites at both the elevations. The other associated species were *Ficus palmata, Pyrus pashia, Toona ciliata, Quercus leucotrichophora* and *Prunus cerasoides* at both elevations. However, *Ficus auriculata, Bauhinia variegata* and *Melia azedarach* were also found at the lower elevation (1000–1400 m) and *Myrica esculenta, Pinus roxburghii, Prunus armeniaca, Rhododendron arboreum, Malus domestica* and *Juglans regia* at the upper elevation (1400–1800 m). At both elevations, the highest number of trees was observed in the 10–20 cm diameter class, followed by 20–30 cm, <10 cm and >30 cm. Total tree carbon stocks (Mg ha$^{-1}$) under the

various diameter classes of the *G. oppositifolia*-based traditional agroforestry system at the different study sites are presented in Table 6. The agroforestry system at both elevations had no significant variation in the total tree carbon stock (TCS). However, among the different sites, the maximum total tree carbon stock was found in Devri and the minimum in Kainchhu, which ranged between 24.23 and 29.16 Mg ha$^{-1}$. Among the elevations, it was recorded as higher in the lower elevation (28.49 Mg ha$^{-1}$) as compared to the upper elevation (25.93 Mg ha$^{-1}$). Total carbon stock in trees of the 10–20 cm diameter class was reported as the highest (15.15 Mg ha$^{-1}$), followed by 20–30 cm (6.99 Mg ha$^{-1}$) and >30 cm (2.75 Mg ha$^{-1}$), and the lowest (2.32 Mg ha$^{-1}$) in the <10 cm diameter class. The species-wise carbon sequestration potentials of the trees under the various diameter classes of the *G. oppositifolia*-based traditional agroforestry system at the different study sites are presented in Table 7. At both the elevations, *G. oppositifolia* reported the highest total carbon stock in all the studied sites under the agroforestry system. At lower elevations, the maximum sequestered amount of carbon was reported for *G. oppositifolia*, followed by *C. australis* at the Pali (2.26 Mg ha$^{-1}$) and Bhali (2.32 Mg ha$^{-1}$) sites, followed by *F. palmata* at the Devri (3.21 Mg ha$^{-1}$) site. At the upper elevation, *G. oppositifolia* is followed by *T. ciliata* (2.63 Mg ha$^{-1}$) at the Moun site, by *Q. leucotrichophora* (3.37 Mg ha$^{-1}$) at the Kotdwara site and by *P. roxburghii* (1.84 Mg ha$^{-1}$) at the Kainchhu site. Among the diameter classes at the lower elevation, the highest total carbon stock was estimated in the 10–20 cm diameter class (18.27 Mg ha$^{-1}$) of *G. oppositifolia* at the Bhali site, while the lowest was estimated in the <10 cm diameter class (0.10 Mg ha$^{-1}$) of *F. palmata* at the Devri site. At the upper elevation, the highest TCS was also estimated in the 10–20 cm diameter class (12.71 Mg ha$^{-1}$) of *G. oppositifolia* at the Moun site, while the lowest was estimated in the <10 cm diameter class (0.03 Mg ha$^{-1}$) of *P. pashia* at the Kainchhu site.

**Table 6.** Carbon stock (Mg ha$^{-1}$) in tree biomass of *G. oppositifolia*-based traditional agroforestry system at two elevations.

| Elevation/Site | | dbh Class | | | | |
|---|---|---|---|---|---|---|
| | | <10 cm | 10–20 cm | 20–30 cm | >30 cm | Total |
| Lower | Devri | 1.13 | 20.83 | 6.96 | 0.24 | 29.16 |
| elevation | Pali | 4.17 | 11.07 | 4.35 | 7.69 | 27.28 |
| (1000–1400 m) | Bhali | 1.96 | 19.49 | 5.22 | 2.36 | 29.03 |
| Upper | Moun | 1.61 | 13.98 | 10.02 | 0.00 | 25.61 |
| elevation | Kotdwara | 2.22 | 12.51 | 7.02 | 6.21 | 27.96 |
| (1400–1800 m) | Kainchhu | 2.84 | 13.01 | 8.38 | 0.00 | 24.23 |
| Avg. | | 2.32 | 15.15 | 6.99 | 2.75 | 27.21 |
| F | | 0.763863 | 0.109047 | 0.103893 | 0.00469 | 1.746479 |
| *p* value | | 0.483155 | 0.897392 | 0.901964 | 0.995322 | 0.208035 |
| F critical | | 3.68232 | 3.68232 | 3.68232 | 3.68232 | 3.68232 |

**Table 7.** Tree carbon stock (Mg ha$^{-1}$) of each tree species in *G. oppositifolia*-based traditional agroforestry system.

| Site | Tree Species | dbh Class | | | | |
|---|---|---|---|---|---|---|
| | | <10 cm | 10–20 cm | 20–30 cm | >30 cm | Total |
| | Lower elevation (1000–1400 m amsl) | | | | | |
| | *Grewia oppositifolia* | 0.90 | 17.97 | 2.19 | – | 21.06 |
| | *Celtis australis* | 0.13 | 0.71 | – | – | 0.84 |
| | *Ficus palmata* | 0.10 | 0.89 | 2.22 | – | 3.21 |
| Devri | *Ficus auriculata* | – | – | 1.83 | – | 1.83 |
| | *Pyrus pashia* | – | 0.43 | 0.72 | – | 1.15 |
| | *Bauhinia variegata* | – | – | – | 0.24 | 0.24 |
| | *Toona ciliata* | – | 0.83 | – | – | 0.83 |

**Table 7.** *Cont.*

| Site | Tree Species | dbh Class | | | | |
|---|---|---|---|---|---|---|
| | | <10 cm | 10–20 cm | 20–30 cm | >30 cm | Total |
| Pali | *Grewia oppositifolia* | 2.80 | 9.69 | 3.88 | 7.27 | 23.64 |
| | *Celtis australis* | 1.37 | – | 0.47 | 0.42 | 2.26 |
| | *Ficus palmata* | – | 0.19 | – | – | 0.19 |
| | *Ficus auriculata* | – | 0.03 | – | – | 0.03 |
| | *Quercus leucotrichophora* | – | 0.99 | – | – | 0.99 |
| | *Melia azedarach* | – | 0.17 | – | – | 0.17 |
| Bhali | *Grewia oppositifolia* | 1.44 | 18.27 | 2.66 | 2.36 | 24.73 |
| | *Celtis australis* | 0.52 | – | 1.80 | – | 2.32 |
| | *Pyrus pashia* | – | 0.59 | 0.76 | – | 1.35 |
| | *Quercus leucotrichophora* | – | 0.40 | – | – | 0.40 |
| | *Prunus cerasoides* | – | 0.23 | – | – | 0.23 |
| Upper elevation (1400–1800 m amsl) | | | | | | |
| Moun | *Grewia oppositifolia* | 1.16 | 12.71 | 5.39 | – | 19.26 |
| | *Celtis australis* | – | – | 0.41 | – | 0.41 |
| | *Ficus palmata* | – | 0.19 | – | – | 0.19 |
| | *Pyrus pashia* | 0.45 | – | 0.37 | – | 0.82 |
| | *Toona ciliata* | – | – | 2.63 | – | 2.63 |
| | *Quercus leucotrichophora* | – | 0.89 | – | – | 0.89 |
| | *Myrica esculenta* | – | 0.19 | – | – | 0.19 |
| | *Pinus roxburghii* | – | – | 0.71 | – | 0.71 |
| | *Prunus armeniaca* | – | – | 0.51 | – | 0.51 |
| Kotdwara | *Grewia oppositifolia* | 1.95 | 11.78 | 4.36 | – | 18.09 |
| | *Celtis australis* | – | 0.21 | – | – | 0.21 |
| | *Ficus palmata* | – | 0.41 | – | – | 0.41 |
| | *Pyrus pashia* | 0.06 | 0.11 | – | – | 0.17 |
| | *Quercus leucotrichophora* | 0.21 | – | 1.07 | 2.09 | 3.37 |
| | *Prunus cerasoides* | – | – | 0.27 | – | 0.27 |
| | *Pinus roxburghii* | – | – | – | 2.81 | 2.81 |
| | *Prunus armeniaca* | – | – | 1.32 | – | 1.32 |
| | *Rhododendron arboreum* | – | – | – | 1.31 | 1.31 |
| Kainchhu | *Grewia oppositifolia* | 2.27 | 11.23 | 5.25 | – | 18.75 |
| | *Celtis australis* | 0.32 | 0.88 | – | – | 1.20 |
| | *Ficus palmata* | – | 0.14 | – | – | 0.14 |
| | *Pyrus pashia* | 0.03 | – | – | – | 0.03 |
| | *Quercus leucotrichophora* | – | 0.37 | 1.29 | – | 1.66 |
| | *Prunus cerasoides* | 0.10 | – | – | – | 0.10 |
| | *Pinus roxburghii* | – | – | 1.84 | – | 1.84 |
| | *Prunus armeniaca* | – | 0.27 | – | – | 0.27 |
| | *Malus domestica* | 0.12 | – | – | – | 0.12 |
| | *Juglans regia* | – | 0.12 | – | – | 0.12 |

## 4. Discussion

The traditional *G. oppositifolia*-based agroforestry systems are of high significance in the Garhwal Himalayan region, providing multiple outputs which generate high income. In the present study, most of the studied parameters of *E. frumentaceae* and *E. coracana* under a traditional agroforestry system show reductions in values as compared to their controls at both the elevations. Higher values under the control treatments for both altitudinal ranges might be due to the absence of trees, as there was no competition for light and nutrients between the crops and trees under the sole cropping system [17,30]. Moreover, in the lower elevation under agroforestry systems, the secretions of chemicals by the bark

and leaves of *G. oppositifolia* and others tree species might have a negative effect on the growth and development of both the crops [31]. The effect of altitudinal gradient and aspect on the production potential of the *G. oppositifolia*-based agroforestry systems has also revealed significant effects, showing the importance of altitude in the selected sites. The results show that for both *E. frumentaceae* and *E. coracana*, greater values for most of the examined growth and yield attributes were reported in lower elevations due to the influence of a higher temperature, which promotes the decomposition rate of leaf litters and improves the physiochemical properties of soil [32]. In addition, the plant population per m$^2$ in both the crops and plant height in *E. frumentaceae* were recorded as higher at the upper elevation range (1400–1800 m amsl), which might be due to the higher germination rate of the seeds, which resulted in the greater number of plants. At the lower elevation, greater values for most of the growth and yield attributes for both crops might also be due to the fact that both *E. frumentaceae* and *E. coracana* are C$_4$ plants, which require warmer growing conditions for better growth and development. The higher elevations are significantly colder in winter as compared to the lower elevations, which decreases the rate of growth and plant development. Higher plant height under the control conditions might also be due to higher plant populations, which resulted in more vertical growth as compared to basal growth of the plants, due to competition for sunlight. Modesto et al. [33] also described that plant height is a morphological variable that is directly related to the crop's population density. Similar findings regarding plant growth and development were also reported by Kausal et al. [31], Kar et al. [34], Kaur et al. [35] and Thakur et al. [36]. The greater number of total and active tillers per plant and higher panicle length under the control conditions might also be due to the impact of the lack of association of tree species, resulting in less competition for available soil nutrients, moisture and sunlight. The growth and development of plants are affected by the absorption of solar radiation and the photosynthetic rate, which are positively correlated with plant growth and yield attributes due to the formation and development of plant organs [37,38]. Bijalwan and Dobriyal [17], Gawali et al. [30], Islam et al. [39], Rahangdale et al. [40] and Kaur and Puri [41] also supported these results. The maximum test weights of *E. frumentaceae* and *E. coracana* were found in the control conditions as compared to the agroforestry system, which might have been due to proper nutrient and water uptake, the direct sunlight received by crops and less competition, which are required for the better growth and development of crops. Similar results were also reported by Chauhan et al. [22], Farhana et al. [42], Kumar and Thakur [43] and Rawat et al. [44]. Longer panicle length and the greater number of active tillers per plant in the control conditions and the lower elevation also resulted in higher grain yield, straw yield and biological yield in both the crops. This could be attributed to changes in the air and soil temperature, soil nutrient availability, water status and high decomposition rate. The yield from crops is also known to be affected by the climatic conditions, edaphic factors and various other biotic and abiotic factors of the selected sites, which prevailed during the cropping period. The results for the various yield parameters are supported by the findings of Chauhan et al. [17], Gawali et al. [30], Islam et al. [39], Kaur et al. [35], Bijalwan [45], Bijalwan et al. [46] and Bijalwan [45]. The harvest index (%) was found to be higher under the agroforestry system for both crops as compared to sole cropping, due to the higher ratio of grain or panicle to shoot in agroforestry systems in comparison with sole cropping systems. Bijalwan [45] and Bijalwan [47] also observed a higher harvest index under various agroforestry systems as compared to the sole cropping system.

Agroforestry has great importance these days due to its multifunctional capacity, including its carbon sequestration potential in different plant species as well as in soil, the socioeconomic aspect and its multiple outputs in farmland [48]. Different agroforestry systems sequester a substantial amount of carbon in plant biomass (including trees, crops and grasses) and soil, which helps to regulate the carbon cycle [49]. In the present study, the tree species associated with *G. oppositifolia* along the altitudinal gradient in traditional agroforestry systems were similar to the findings of Kumar et al. [50]. The results show that

*G. oppositifolia* is a dominant agroforestry tree species at both elevations. *G. oppositifolia* was found to be associated with *Celtis australis*, *Ficus palmata*, *Pyrus pashia*, *Prunus cerasoides* and *Quercus leucotrichophora* at the lower elevation, and along with these species, *Pinus roxburghii* and *Prunus armeniaca* were also found in all the sites of the higher elevation. In the various agroforestry systems of the Tehri district of Garhwal Himalaya, several tree species such as *Grewia oppositifolia*, *Melia azedarach*, *Celtis australis*, *Toona ciliata*, *Quercus leucotrichophora*, *Prunus cerasoides*, *Juglans regia*, *Rhododendron arboreum*, *Bauhinia variegata*, *Ficus palmata*, *Ficus auriculata* and *Pyrus pashia* are found between 1200 and 2000 m amsl [51,52]. Similar species were also reported by Manzoor and Jazib [53] in the agroforestry system of Jammu and Kashmir, India. At both elevations, higher numbers of trees were present in the 10–20 cm diameter class and numbers of tree species decreased with the increasing diameter class, which might be due to the reducing pattern of agricultural practices, the removal of older trees from agricultural fields to reduce the shading effect and fuelwood collection. The highest amount of tree carbon stock was reported in the 10–20 cm diameter class, which is due to the greater number of trees in this diameter class, and the lowest carbon stock in the >30 cm diameter class was because the lowest number of trees were in this diameter class. According to Nero et al. [54], in an ecosystem, the trees in the lower diameter class had the highest tree densities and species diversity, which is directly correlated with the tree carbon sequestration potential. A similar result was also obtained by Hauchhum [55], who reported that the maximum number of trees were observed in the 20–30 cm and 30–40 cm diameter class (lower diameter class) as compared to the 40–50 cm diameter class. Bijalwan and Dobriyal [37] also reported that an agrisilviculture system at an elevation range of 1000–1500 m on the northern aspect recorded the highest number of trees under the 10–20 cm diameter class, while the lowest number of trees were recorded under the 50–60 cm diameter class, which supports the findings of the present study. The carbon sequestration potential of traditional agroforestry systems is reliant on the density, diameter, age and structure of trees, and thus variation in carbon stock at the different sites was reported. Increased tree density, basal area and frequent diameter class distribution with decreasing altitude may explain the increased total carbon stock in the present study at lower elevations. These results are also in agreement with the observations recorded by Rana et al. [38], who reported that carbon density is significantly influenced by elevation. The study also recorded that the rate of carbon sequestration decreases with an increase in altitude, which could be due to the temperature, and consequently its impact on the plant metabolism, superior tree diameters, tree density and basal area. Coomes and Allen [56] also reported that with increasing altitude the growth of trees declines and biomass per tree increases with increasing diameter. Kumar et al. [50] and Bijalwan et al. [57] also estimated the maximum amount of carbon in lower elevations as compared to middle and higher elevations, under the traditional agroforestry systems of Indian Himalayan regions. Bijalwan et al. [58] also observed that in an agrisilviculture system, *G. oppositifolia* captured maximum carbon (11.17 Mg ha$^{-1}$) at 1000–1500 m elevation, which was lower than the value of the present study. Similar studies have also been conducted by Kumar and Thakur [43] and Bijalwan et al. [57]. Among the species, maximum tree carbon stock was reported in *G. oppositifolia*, followed by *Celtis australis* and *Ficus* spp., which might be due to the dominance of these tree species in the lower and upper elevation of the studied sites. Similar results were reported by Kumar et al. [52] under a *Quercus leucotrichophora*-based agroforestry system in Central Himalaya, India. Vikrant et al. [59] also reported a significant contribution by *G. oppositifolia* in the carbon stock contributed by different tree species in the agroforestry systems of the Tehri district of Uttarakhand, India. Agroforestry is an integral part of the ecosystem in the Garhwal Himalayan region, providing multiple benefits. Various tree species, such as *G. oppositifolia*, are abundant in these agroforestry systems and have a significant potential for combating climate change, while also enhancing farmer income.

## 5. Conclusions

Based on the findings of the present study, it is concluded that the reduction in the grain yield of *Echinochloa frumentacea* and *Eleusine coracana* under the *Grewia oppositifolia*-based agroforestry system at both elevations is not very large as compared to the sole cropping system. Among the studied elevation ranges, *G. oppositifolia* was found to be associated with *Celtis australis*, *Ficus palmata*, *Pyrus pashia*, *Quercus leucotrichophora*, *Prunus cerasoides*, *Pinus roxburghii* and *Prunus armeniaca*. *G. oppositifolia* and associated tree species also store a significant amount of carbon in tree biomass at both elevation ranges (i.e., 28.49 Mg ha$^{-1}$ and 25.93 Mg ha$^{-1}$ at the lower and upper elevations, respectively), which helps in the mitigation of atmospheric carbon and global warming. The *G. oppositifolia* tree is observed to be the most preferred tree by the farmers and also provides green fodder during lean periods. Moreover, the cultivation of agricultural crops under an agroforestry system is a tradition which provide nutritious food grains and straw for animals; thus, farmers prefer to adopt the *G. oppositifolia*-based traditional agroforestry system on their farmlands.

**Author Contributions:** Conceptualization, N.T. and A.B.; methodology, A.B. and S.C.; software, S.T., S.K. and T.K.T.; validation, N.T., A.B., S.C. and S.K.; formal analysis, N.T., S.T. and S.K.; investigation, N.T.; resources, N.T. and A.B.; data curation, N.T.; writing—original draft preparation, N.T., S.T. and S.K.; writing—review and editing, N.T., A.B., S.C., B.S., C.S.D., S.T., M.K., S.K., M.M.S.C.P. and T.K.T.; visualization, A.B. and T.K.T.; supervision, A.B., S.C., B.S., C.S.D. and S.K.; project administration, A.B. All authors have read and agreed to the published version of the manuscript.

**Funding:** This research received no external funding.

**Institutional Review Board Statement:** Not applicable.

**Informed Consent Statement:** Not applicable.

**Data Availability Statement:** The data presented in this study are available on request from the corresponding author.

**Conflicts of Interest:** The authors declare no conflict of interest.

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
