# Peer review of "Crop Production and Carbon Sequestration Potential of Grewia oppositifolia-Based Traditional Agroforestry Systems in Indian Himalayan Region"

_land, doi:10.3390/land11060839_

Round 1

Reviewer 1 Report

LAND document 1726564 “Crop production and carbon sequestration potential of traditional agroforestry systems based on Grewia oppositifolia in the Indian Himalayan region” has technical-scientific merit and is a technology with high potential for adoption for sustainable agriculture in the Indian Himalayan region.

However, it needs some changes and clarifications listed below:

- What is the explanation for the lower harvest in the agroforestry system and its higher harvest index (AI)?

- I consider it important to provide information on the average population of trees at each of the altitudes evaluated, as this factor can influence the results obtained. You can randomly sample and display the population of trees at each altitude.

- Why was the sampling and analysis of C in the soil not carried out? To determine C sequestration, the role of soil must also be considered. Or is it irrelevant in this condition?

- The conclusion contradicts the results, except for the last sentence which I consider suitable to be kept as a conclusion.

Therefore, considering the technical-scientific relevance, the innovation aspects and the importance of this information for sustainable agriculture in the Indian Himalayan region, I recommend this article after including the suggestions and changes indicated by the review.

Author Response

  1. What is the explanation for the lower harvest in the agroforestry system and its higher harvest index (AI)?

Response 1: Harvest Index is a relationship of grain and straw ratio. Harvest Index depending on grain (or economical) yield and biological yield (grain yield + straw yield).  The HI (ratio of grain to straw/shoot) in the studied agroforestry systems is recorded higher than sole cropping systems. The higher value to harvest index represents the less effect of tree crops interaction on grain production of both millet crops under Grewia oppositifolia based agroforestry.

  1. I consider it important to provide information on the average population of trees at each of the altitudes evaluated, as this factor can influence the results obtained. You can randomly sample and display the population of trees at each altitude.

Response 2: We have selected two elevation (1000-1400 m amsl and 1400-1800 m amsl) based on the pilot survey because Grewia oppositifolia based agroforestry is mainly distributed between these elevations in the study area. After the selection of study area based on the availability of Grewia oppositifolia based agroforestry system and use, we adopt Stratified Random Sampling approach for data collection. 

  1. Why was the sampling and analysis of C in the soil not carried out? To determine C sequestration, the role of soil must also be considered. Or is it irrelevant in this condition?

Response 3: Soil ecosystem play a significant role in role in carbon sequestration, however, in the present study we focused on change in growth and yield attributes of crop with Grewia oppositifolia tree and carbon stock mainly.

  1. The conclusion contradicts the results, except for the last sentence which I consider suitable to be kept as a conclusion.

Response 4: The conclusion of the article has modified in accordance of experimental results.

Reviewer 2 Report

The paper entitled “Crop production and carbon sequestration potential of Grewia oppositifolia based traditional agroforestry systems in Indian Himalayan region” reflects the development of applied research, the topic is interesting and the manuscript has an approach innovative. However, the methods are not fully described, introduction, results and discussion must to be improved. Thus, major changes are recommended.

Comments

1) Introduction needs structuring. Also, the literature review is focused on literature from India, and for example for the definition of agroforestry there are best suited definition worldwide (see for example Agroforestry systems journal).

2) Introduction – when the authors refer country it is India?

3) Page 2, 2nd paragraph – this paragraph need structuring and an English revision.

4) scientific names of the species should be in italics.

5) Figure 1 is not sited in the text.

6) Coordinate system is missing.

7) Section 2 Material and Method need to be rewritten, the description of the materials and methods has not a good sequence and there is information missing regarding. Also, references to the data used and methods are missing.

8) what do the authors mean by 2.4 Observation (for agricultural crops)?

9) The plots are pure or mixed? The subsequent analysis of the plots depends on its composition as the interaction between trees in pure and mixed plots is different.

10)What is above ground biomass density?

11) Figure 2 should be improved. The scale should be presented as scale bar. It is not correct to present scale with bar and text. Also, it is very difficult to read the text in the right inferior figure. If the authors add a letter to each figure as well as in the figure caption, it would improve its understanding.

12) Results were difficult to evaluate as the materials and methods were not fully described. Structuring is needed. 13) Results – there are some repetitions. The same information is in the text, tables and figures. Please revise.

14) Discussion – the authors should discuss only their results and compare it with published references.

Author Response

Attached details rebuttal comments.

Round 2

Reviewer 2 Report

The paper entitled “Crop production and carbon sequestration potential of Grewia oppositifolia based traditional agroforestry systems in Indian Himalayan region” has improved in the second version of the manuscript. However there are still some issues that should be addressed. Thus, it is recommended to accept the manuscript after minor changes.

Comments

1) Please avoid using etc..

2) What do the authirs mean by “Grewia based, Celtis based, Quercus based etc.”?

3) 2.2. Climate and Soil – References are missing. Where did the authors get the data from?

4) Figure 2 should be improved. The scale should be presented as scale bar. It is not correct to present scale with bar and text. Also, it is very difficult to read the text in the right inferior figure. If the authors add a letter to each figure as well as in the figure caption, it would improve its understanding.

5) “Tree volumes and carbon stocks were measured”. Volume and biomass is estimated not measured.
